# Small Extracellular Vesicles Harboring PD-L1 in Obstructive Sleep Apnea

**DOI:** 10.3390/ijms25063208

**Published:** 2024-03-11

**Authors:** Sylvain Recoquillon, Sakina Ali, Grégoire Justeau, Jérémie Riou, M. Carmen Martinez, Ramaroson Andriantsitohaina, Frédéric Gagnadoux, Wojciech Trzepizur

**Affiliations:** 1SFR ICAT, Team Carme, MitoVasc Laboratory, UMR CNRS 6015 INSERM 1083, University of Angers, 49000 Angers, France; sylvain.recoquillon@univ-angers.fr (S.R.); frgagnadoux@chu-angers.fr (F.G.); 2INSERM 1063, University of Angers, 49045 Angers, France; sakina.binti.ali@gmail.com; 3Department of Respiratory and Sleep Medicine, Angers University Hospital, 49100 Angers, France; gregoire.justeau@gmail.com; 4Delegation for Clinical Research and Innovation, Angers University Hospital, 49100 Angers, France; jeremie.riou@univ-angers.fr; 5PhyMedExp, Montpellier University, INSERM, CNRS, CHRU Montpellier, 34295 Montpellier, France; carmen.martinez@inserm.fr (M.C.M.); ramaroson.andriantsitohaina@inserm.fr (R.A.)

**Keywords:** sleep disordered breathing, cancer, exosomes, immune checkpoint

## Abstract

Obstructive sleep apnea syndrome (OSA) has been associated with increased cancer incidence and aggressiveness. One hypothesis to support this association is the implication of immune response, particularly the programmed cell death pathway, formed by the receptor PD-1 and its ligand PD-L1. Recent studies have shown dysregulation of this pathway in severe OSA patients. It has also been shown that small extracellular vesicles (sEVs) carrying PD-L1 induce lymphocyte dysfunction. Thus, the aim of our study was to analyze the expression of PD-L1 on sEVs of OSA patients and to evaluate the role of sEVs on lymphocyte activation and cytotoxicity. Circulating sEVs were isolated from OSA patients and the control group. Lymphocytes were isolated from the control group. Circulating sEVs were characterized by western blot, nanotracking analysis, and flow cytometry and were incubated with lymphocytes. Our results show no differences in the quantity and composition of sEVs in OSA patients and no significant effects of sEVs in OSA patients on lymphocyte activation and cytotoxicity. These results suggest that OSA does not modify PD-L1 expression on sEVs, which does not contribute to dysregulation of cytotoxic lymphocytes.

## 1. Introduction

With nearly one billion adults affected worldwide and a prevalence of 13% in men and 6% in women, obstructive sleep apnea (OSA) syndrome is a highly prevalent disease characterized by recurrent episodes of complete or partial upper airway obstruction during sleep [1]. The direct consequences of OSA are sleep fragmentation with microarousals, hypoxia–reoxygenation cycles, cardiac frequency and systemic arterials, and transient increases in pulmonary pressure.

Recent studies have demonstrated that patients with severe OSA, especially those with significant nocturnal hypoxemia, have a higher risk of cancer incidence and mortality [2,3,4,5,6], which is not modified by adherent positive airway pressure therapy [7]. Cancer development could be favored by hypoxia–reoxygenation cycles. This hypothesis is supported by experimental data from both in vitro and in vivo experiments. Indeed, intermittent hypoxia (IH), an experimental model used to reproduce OSA hypoxia–reoxygenation cycles, increase as tumor aggressiveness through a higher proliferation of cancer cells as well as invasiveness ability [8,9]. Other studies suggest that OSA could favor tumor development by disturbing the immune response through IH [10].

Immune response is a complex phenomenon based on a balance between activation and inhibition of cells composing the immune system, notably T cells. This balance could be destabilized in cancer, leading to inhibition of the immune response and contributing to the disease’s progression. One of the major inhibitory pathways of T cells is the programmed cell death pathway, composed of the receptor programmed cell death 1 (PD-1) and its ligand programmed death-ligand 1 (PD-L1). This pathway plays an important role in T cell quiescence maintenance [11] and in the immune tolerance of the organism against cancer cells. The PD-1 ligands, PD-L1 and PD-L2, when expressed on tumor cells, act as an inhibitory signal on T cells, reducing the cytotoxicity of these cells and favoring tumor development. Beyond the effect of this pathway on tumor development, high levels of soluble PD-L1 and PD-1 are associated with poor prognosis and could be used as markers in various cancers, such as colorectal [12], pancreatic [13], or non-small cell lung cancer [14]. Currently, antibodies directed against PD-1 and PD-L1 are now part of the therapeutic arsenal to fight against numerous types of cancer, including lung cancer.

OSA, through IH, could activate the PD-1/PD-L1 pathway, favoring tumor development. In fact, in OSAS patients, higher expressions of PD-1 and PD-L1 have been found in CD8 T cells and monocytes, respectively, independent of any cancer pathology [10]. A recent study suggests that circulating small extracellular vesicles (sEVs) from head and neck cancer patients could carry PD-1 and PD-L1, and these sEVs could inhibit the activity of T cells in vitro [15].

sEVs are nanoscale vesicles (30–100 nm) produced by numerous cells, including vascular cells, circulating cells, and cancer cells [16]. sEVs’ composition reflects the cells from which they originate. These vesicles are implicated in intercellular communication by carrying genetic material, proteins, and lipids [17] and may act at different stages of cancer development: angiogenesis, acquisition of resistance to treatment, tumor proliferation, or during the modulation of the immune response [18,19]. sEVs from OSA patients increase the malignancy of tumor cells in in vitro and in vivo models [20]. sEVs have also been studied as diagnosis and prognosis biomarkers in non-small cell lung cancers [21,22].

Currently, the composition and potential impact of circulating sEVs from OSA on immune response modulation are not established. The objective of this study is to analyze the composition of sEVs in moderate-to-severe OSA patients, especially in terms of markers of cellular origin and PD-L1 expression, and to assess the effect of sEVs on CD8 lymphocyte activation and cytotoxicity.

## 2. Results

### 2.1. Patients’ Characteristics

Ninety patients were included in this study. Their baseline characteristics are summarized in Table 1. Patients were split into the control (AHI < 15/h) and moderate-to-severe OSA groups (AHI ≥ 15/h) according to their AHI (median (IQR) AHI 5.00 (2.50–8.00) for the control group versus 29.0 (22.00–43.00) for the moderate-to-severe OSA group; *p* < 0.001). Patients and controls differed by age (median (IQR) age 42.00 (36.50–55.50) years for controls versus 58.00 (51.00–66.00) years for moderate-to-severe OSA, *p* < 0.001), body mass index (BMI) (median (IQR) BMI 25.60 (23.50–29.05) kg.m^2^ for controls versus 31.10 (27.30–35.00) kg/m^−2^ for moderate-to-severe OSA, *p* < 0.001), and blood pressure. Control patients had lower levels of fasting glucose and LDL cholesterol as compared to moderate-to-severe OSA. No differences were observed between the two groups regarding the white blood cell count (neutrophils, lymphocytes, and monocytes).

### 2.2. Characterization of Plasma sEVs

To assess the quality of sEV isolation, we performed a western blot to identify markers of sEVs. As shown in Figure 1A and as expected, we observed sEV markers (CD81, CD9, TSG101, and ALIX) on the sEV samples. In contrast, we did not depict any sEV markers, neither on large extracellular vesicles (lEVs) nor in cell lysates. On the lEVs and cell lysates, we observed expected markers such as α-tubulin, β-actin, and GAPDH, which were not detectable in the sEV samples. The size of the particles isolated was determined by NTA analysis. Figure 1B shows examples of histograms obtained from samples of control and moderate-to-severe OSA patients. The median size of isolated particles corresponded to the expected size of sEVs, less than 100nm. No differences in sEV mean size were observed between the two groups (Figure 1C). Figure 1D shows the characteristic sEVs round shape, delimited by a lipid bilayer.

### 2.3. Effect of OSA on Plasma sEV Concentration and Cellular Origin

As shown in Figure 2A, we did not observe any difference in plasma sEV concentration as measured with NTA between the two groups. We performed the analysis of 37 surface epitopes of sEVs by flow cytometry to determine the expression of the sEV markers and their cellular origin (Figure 2B). As expected, the expression of characteristic sEV markers (CD9, CD63, and CD81) was confirmed. Furthermore, the highest signals were obtained on CD42a, CD62P, CD41a (platelet markers), HLA-DPDQDR (HLA-class II), and CD29 (leucocyte markers) beads. This result suggests that platelets and leucocytes are the main sources of sEVs isolated from plasma. However, the expression of sEV surface markers was not significantly different between the groups. We further evaluated the expression of PD-L1 by western blot (Figure 2C). As shown in Figure 2C, we did not highlight any differences in PD-L1 expression between control and moderate-to-severe OSA patients. Furthermore, no correlation was found between AHI and sEVs PD-L1 expression (Figure 2D). No additional differences were observed using the AHI 30/h threshold (median (IQR) PD-L1/total protein 7.00 (2.60–8.85) for the AHI < 30/h group versus 5.8 (4.15–7.60) for the severe OSA group; *p* > 0.05) or separating patients by median T90 (median (IQR) PD-L1/total protein 8.00 (5.97–10.32) for the T90 ≤ 9% group versus 6.0 (4.65–7.80) for the T90 > 9% group; *p* > 0.05).

### 2.4. Effect of sEVs on CD8 Lymphocyte Activation and Cytotoxicity

To determine whether sEVs from moderate-to-severe OSA patients could modify the activity of CD8 lymphocytes in vitro, we isolated CD8 lymphocytes from the control group. After incubation with sEVs, we stimulated these lymphocytes with activation beads for 24 h and analyzed the lymphocyte activation (CD69 labeling) and cytotoxicity (perforin labeling). As shown in Figure 3A and 3B, the activation beads increased the CD69 and perforin fluorescence intensity. When we stimulated lymphocytes with sEVs from either control or moderate-to-severe OSA patients alone, no changes in CD69 and perforin fluorescence intensities were observed. Moreover, neither sEVs from control nor from moderate-to-severe OSA modified the increase in CD69 and perforin fluorescence intensity induced by activation beads. This result suggests that sEVs isolated from moderate-to-severe OSA patients did not play a role in the activation and cytotoxicity of CD8 lymphocytes in vitro under this experimental condition.

## 3. Discussion

The present study reports that moderate-to-severe OSA did not modify the sEVs in terms of concentration, size, cellular origin, or PD-L1 expression. The sEVs from OSA patients did not affect lymphocyte activation or cytotoxicity in vitro. Thus, PD-L1 expressed on sEVs is not a valuable marker or target for reduced immune tolerance in OSAs participating in lung cancer aggressiveness.

The role of EVs, and particularly sEVs, is now a field of biology with wide interest. Indeed, in the last decade, an impressive number of studies have been published describing their characterization, composition, and impact on several diseases, such as cardiovascular diseases and cancers [23]. In OSA, some studies evaluated sEV miRNA content and showed their impact on endothelial cells and monocyte/macrophage function [24].

Circulating levels of sEVs might be affected by different pathological conditions. An in vitro study showed that cancer cells can produce up to 25,000 vesicles per cell per 48 h [25], which could contribute to a global increase in the circulating level. Some authors have raised the possibility that tumor-derived sEVs could represent 10% of the total circulating sEVs [26]. In colorectal cancer, a higher level of circulating sEVs was found in patients as compared to controls, and this increase was associated with a decrease in overall survival, suggesting that sEVs could represent a prognostic biomarker [27]. Circulating sEVs obtained from obesity hypoventilation syndrome (OHS) patients enhance lung adenocarcinoma tumor cell line proliferation, migration, and invasion, and these effects are at least partially reversible with adherent continuous positive airway pressure treatment [28]. However, the specific sEVs cargos in OHS patients and their responses to treatment, as well as murine models targeting some pathways through selective alterations of sEVs miRNA cargo, have not been conducted. Currently, the effect of OSA on plasma sEV concentration is not well established. Limited studies compared the total concentration of plasma sEVs in non-OSA and OSA patients and showed no differences [29]. Furthermore, continuous positive airway pressure treatment did not impact sEV concentration in severe OSA patients [30,31]. The current results confirmed the absence of a significant impact of OSA on total plasma sEV concentration.

To the best of our knowledge, this is the first study evaluating the cellular origin of sEVs in OSA patients. To date, such analysis has only been performed in various experimental models of OSA, including IH and sleep fragmentation. In vitro studies have shown that IH increases the release of sEVs by endothelial cells or cardiomyocytes [32,33]. Plasma sEVs from endothelial progenitor cells, platelets, and monocytes were increased in mice exposed to sleep fragmentation [8]. Similar changes were found in healthy volunteers exposed to four days of IH [34]. In the present study, we measured 37 surface markers of sEVs. The most represented sEV population are platelet sEVs and, to a lesser extent, sEVs of leucocyte origin. No significant differences were found between the two groups of patients.

An implication of sEVs on tumor malignant properties has been previously suggested in OSA. Thus, sEVs isolated from OSA patients and from different OSA in vivo and in vitro experimental models increased cancer cell proliferation, migration, and invasion capacities [20,33,35]. Furthermore, sEVs induce endothelial barrier disruption and enhance extravasation properties. In the present study, we aimed to further investigate the implication of sEVs on oncogenesis in OSA by investigating their role in PD-1/PD-L1-mediated lymphocyte activity. No differences in PD-L1 expression nor any implication of sEVs from OSA patients in in vitro lymphocytes’ activation and cytotoxicity have been observed. The PD-L1/PD-1 pathway/OSA association was first described on lymphocytes and monocytes by Cubillos-Zapata and colleagues. Two years later, further analysis from the same team demonstrated that this association was solely present in severe OSA patients up to 55 years of age [35]. In our study, the median age of the patients was 57.82 years in the moderate-to-severe OSA. Consequently, further analysis was performed in the subgroup of patients up to 55 years of age but did not show any difference regarding PD-L1 expression.

Some limitations should be considered when interpreting the findings of our study. First, the allocation of patients to the control or moderate-to-severe OSA group was based on the AHI. Since the initial conception of the study, it has been shown that the time spent under 90% saturation (T90), in contrast to AHI, is the major independent predictor of incident cancer [4]. Consequently, we also separated patients based on the T90. However, this post hoc analysis did not reveal any significant differences between groups. Secondly, we did not include any healthy subjects in the control group and allocated mild OSA patients to the control group. This choice was motivated by the absence of any significant increase in cancer incidence or aggressiveness demonstrated in patients with an AHI < 15/h [3]. Furthermore, in the Princeps publication on the OSA/PD-L1 association, a significant increase was observed solely in severe OSA patients [10]. The absence of any trend in the association between sEV expression and AHI (Figure 2D) suggests that different cut-offs of AHI would not impact the major result.

## 4. Materials and Methods

### 4.1. Patients

Patients were recruited at the Department of Respiratory and Sleep Medicine in the Hospital of Angers. All patients provided their informed consent to participate in the study. Men and women aged more than 18 years old with a clinical suspicion of OSA were included in the study. The exclusion criteria were a prior treatment for OSA and a previous history of cancer. Patients with no OSA or mild OSA (AHI < 15/h; control group) were compared to patients with moderate-to-severe OSA (AHI ≥ 15/h).

### 4.2. Sleep Recordings

Patients were evaluated using laboratory polysomnography (PSG) with the following criteria: Apnea is defined as at least a 90% decrease in the oronasal sensor signal, and hypopnea is defined as at least a 30% decrease in the nasal pressure signal combined with either ≥ 3% arterial oxygen desaturation or an arousal (PSG), both lasting at least 10 s [36]. Certified sleep physicians read the PSG data and confirmed the absence or presence of OSA and its severity.

### 4.3. Blood Sampling

Blood samples were taken in the morning following polysomnography after overnight fasting from a peripheral vein using a 21-gauge needle to minimize platelet activation and collected in EDTA tubes (Vacutainers, Becton Dickinson, Le Pont de Claix, France).

### 4.4. Isolation of Circulating sEVs

Blood was centrifuged at 250× *g* for 20 min. Plasma-rich platelets were harvested and centrifuged at 1500× *g* for 20 min. The plasma-poor platelet was then harvested and centrifuged at 17,000× *g* for 45 min to pellet lEVs. The supernatant, containing the sEVs, was then ultracentrifuged at 100,000× *g* for 1 h to isolate sEVs. The sEVs’ pellet was washed with 1× PBS with a last ultracentrifugation at 100,000× *g* for 1 h. The pellet was finally resuspended in 100 µL of NaCl and conserved at 4 °C until analysis.

### 4.5. Transmission Electronic Microscopy

For electronic microscopy, 5 µL of sample containing exosomes were mixed with 5% glutaraldehyde in phosphate buffer and deposited on a formwar-carbon grid (FCF150-CU-SB, EMS, Hartfield, PA, USA). The fixation step lasted 15 min at room temperature. The grid was then rinsed for 90 s with distilled water. This step has been repeated 6 times. After the last rinse, the grid was immediately deposited on a drop of uranyl acetate (0.4%) and methylcellulose (2%) for 10 min on ice and in the dark. The excess of uranyl acetate/methylcellulose was gently removed by blotting on filter paper. The grids were observed on a TEM JEM-1400 (Jeol, Tokyo, Japan) at 120 kV. The images were acquired with an Orius 832 Camero from Gatan Inc. (Pleasanton, CA, USA).

### 4.6. Analysis of Plasma sEV Concentration and Size by Nanoparticle Tracking Analysis (NTA)

The sEVs samples were diluted in sterile NaCl before acquisition on a NanoSight NS3000 equipped with a 405nm laser (Malvern Instruments Ltd., Worcestershire, UK). For each sample, 5 videos have been recorded with an automatically monitored temperature range of 20 °C to 21 °C. Videos were then analyzed when sufficient and valid vesicle trajectories were measured. Concentrations of sEVs have been normalized by the collected plasma volume for each patient. Data capture and further analysis have been performed using the NTA software version 3.1.

### 4.7. Phenotyping of sEVs Using the MACSPlex Exosomes Kit

To evaluate surface markers of sEVs, we used the commercial MACSPlex Exosome kit (Miltenyi Biotec, Bergisch Gladbach, Germany). This kit allows for the analysis of 37 different surface markers on sEVs [37]. For each sample, 4 µg of sEVs were used, and the experiment was performed according to the manufacturer protocol. Briefly, 120 µL of sEVs samples were added to the wells of a MACSPlex 96-well 0.22 µm filter plate, and 15 µL of MACSPlex Exosome Capture Beads were added. Filter plates were then incubated overnight at room temperature on an orbital shaker. The complex composed of the beads and exosomes was washed with 200 µL MACSPlex buffer, and the liquid was removed by centrifuging 300× *g* for 3 min at room temperature. A mix composed of 5 µL of APC-conjugated anti-CD9, anti-CD63, and anti-CD81 detection antibodies was added to each well, and the plate was incubated for 1 h at room temperature on an orbital shaker. The samples were washed by centrifugation at 300× *g* for 3 min at room temperature, resuspended in MACSPlex buffer, and analyzed by flow cytometry with a MACSQuant Analyzer 10 flow cytometer (Miltenyi Biotec).

### 4.8. Western Blot

Western blot was used to characterize sEVs and study the expression of PD-L1 on the sEVs. Briefly, 5 µg of sEVs proteins were separated by 4–20% gel electrophoresis (Bio-Rad, Hercules, CA, USA). Proteins were then transferred to the nitrocellulose membrane, which was blocked with 5% bovine serum albumin (BSA) in TBS-T 1X. The membranes were probed with mouse antibodies for CD9 (1:1000, Santa Cruz Biotechnology, Dallas, TX, USA), CD81 (1:1000, Santa Cruz Biotechnology), ALIX (1:250, BioLegend, San Diego, CA, USA), GAPDH (1:1000, Abcam, Cambridge, UK), β-actin (1:5000, Sigma-Aldrich, Saint Quentin Fallavier, France), goat antibodies for α-tubulin (1:1000, Santa Cruz Biotechnology), PD-L1 (1:1000, R&D Systems, Minneapolis, MN, USA), and rabbit antibodies for TSG101 (1:1000, Santa Cruz Biotechnology). After washing, bound antibodies were detected with a secondary peroxidase-conjugated anti-mouse (1:5000, Sigma-Aldrich), anti-goat antibody (1:5000, US Biological, Salem, MA, USA), or anti-rabbit (1:20,000, US Biological). The bands were visualized using the enhanced chemiluminescence system and quantified by densitometry using Image J (version 1.52a).

### 4.9. Isolation of CD8 Lymphocytes

CD8^+^ lymphocytes were isolated from the control group patients’ blood. Blood was loaded on the density gradient Histopaque-1077 (Sigma-Aldrich) and centrifuged at 400× *g* for 30 min. This allows for the isolation of peripheral blood mononuclear cells (PBMCs). Then, PBMCs were washed once with PBS and incubated for 5 min at room temperature with 3 mL of ACK lysis buffer (150 nM NH_4_Cl, 10 mM KHCO_3_, 0.1 mM Na_2_EDTA, pH 7.2–7.4) to lyse contaminants in red blood cells. After adding 12 mL of PBS, the preparation was centrifuged at 800× *g* for 10 min. The supernatant was discarded, and the pellet corresponding to the PBMCs was resuspended in 10 mL of PBS. PBMCs were centrifuged at 800× *g* for 10 min. The supernatant was discarded, and the pellet was resuspended in PBS supplemented with 0.5% of BSA and 2 mM EDTA. PBMCs were then incubated with a magnetic-labeled antibody for CD8 (Miltenyi Biotec) for 30 min at 4 °C. After incubation, CD8^+^ cells were separated using a selection column placed on a magnet. CD8 lymphocytes were then cultured for one day in culture medium RPMI-1640 (Sigma-Aldrich) supplemented with 10% fetal bovine serum and 1% antibiotics (Sigma-Aldrich) before stimulation.

### 4.10. Analysis of CD8^+^ Lymphocyte Activation and Cytotoxicity

To activate CD8 T lymphocytes, activation beads were used according to the manufacturer’s protocol (Miltenyi Biotec). These beads consist of anti-biotin beads labeled with biotinylated antibodies directed against CD2, CD3, and CD28. These antibody-labeled beads simulate antigen-presenting cells to activate T lymphocytes. After their isolation, CD8 T lymphocytes were incubated with sEVs from control and OSA patients for 1 h at the plasma concentration of each patient. The lymphocytes were then stimulated with one activation bead for two CD8 T lymphocytes (bead-to-cell ratio 1:2). After 24 h of stimulation, cells were incubated with fluorescent antibodies for CD69 and perforin (Miltenyi Biotec) for 30 min at 4 °C. After incubation, fluorescence was analyzed by a flow cytometer 500 MPL system (Beckman Coulter, Villepinte, France) with the MXP analysis software version 2.2 (Beckman Coulter).

### 4.11. Statistical Analysis

Quantitative variables were reported as mean ± standard deviation (SD) when their distribution could be considered Gaussian, and with medians and interquartile ranges (IQR) otherwise. Comparisons between groups based on AHI (no OSA or mild OSA [AHI < 15/h; control group] versus moderate-to-severe OSA [AHI ≥ 15/h]) were performed using Student or Mann-Whitney tests. The following data were successively compared: clinical, PSG, and biological characteristics at inclusion; exosome size and concentration; exosome PD-L1 expression; lymphocyte CD69 and perforin expression in the presence of activating beads and exosomes. *p* < 0.05 was considered to be statistically significant.

Based on a recent study considering an average expression of PD-L1 of about 25% on monocytes from patients with AHI < 15/h and 40% on monocytes from patients with AHI ≥ 15/h, with a standard deviation in both groups of 20%, 45 patients per group were needed to achieve 90% power using a type-I error rate of 5%.

Statistical analyses were performed using R software (version 3.5.1, R-Core Team, Vienna, Austria).

## 5. Conclusions

The present study does not underscore any significant impact of sEVs from OSA patients on T-cell proliferation and cytotoxic activity. The previously demonstrated overexpression of PD-L1 in OSA patients on lymphocytes was not observed on sEVS. Overall, these findings do not support the hypothesis of a potential biological role for sEVs acting on the PD-L1 pathway in OSA-associated immune dysregulation.

## Figures and Tables

**Figure 1 ijms-25-03208-f001:**
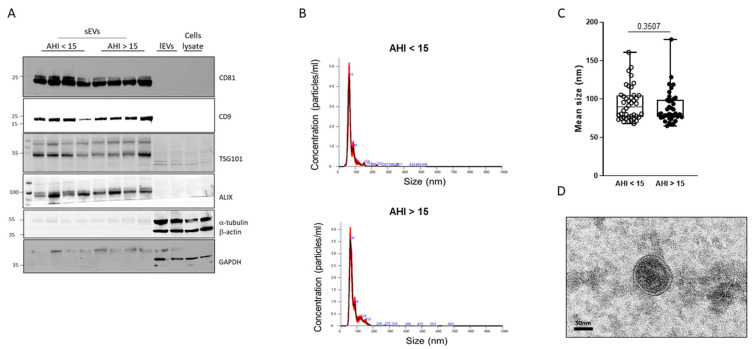
Characterization of circulating sEVs isolated from blood. (**A**) Western blot analysis of sEV markers CD81, CD9, TSG101, and ALIX and lEV markers (β-actin, α-tubulin, and GAPDH). (**B**) Examples of curves obtained by nanotracking analysis for circulating sEVs isolated from control (AHI < 15, upper panel) and moderate-to-severe OSA (AHI > 15, lower panel) patients. The black line represents the mean concentration of particles in function of their size, and red color represents the standard deviation of the concentration for each value of size. (**C**) Analysis of circulating sEV median size in control (AHI < 15) and moderate-to-severe OSA (AHI > 15) patients. (**D**) Transmission electronic microscopy observation of plasma exosomes from patients.

**Figure 2 ijms-25-03208-f002:**
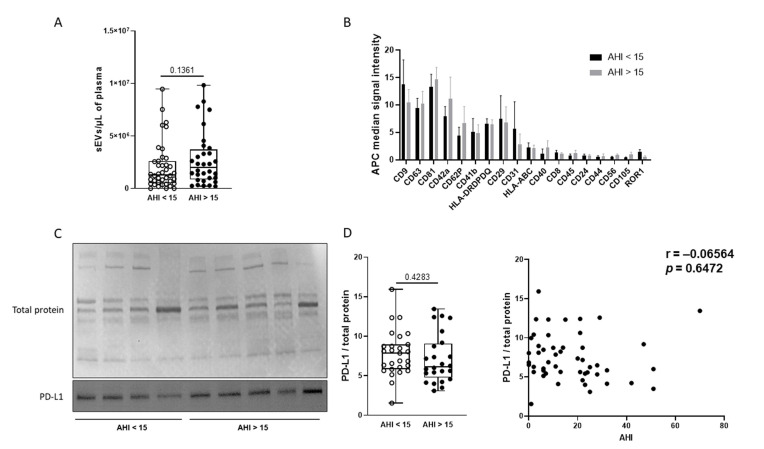
Composition of circulating sEVs is similar between control and moderate-to-severe OSA patients. (**A**) Concentration of circulating sEVs analyzed by NTA in control (AHI < 15) and moderate-to-severe OSA (AHI > 15) patients. (**B**) Cellular origins studied by surface markers of circulating sEVs and determined by flow cytometry using the MACSplex exosomes kit (Miltenyi Biotec, Bergisch Gladbach, Germany) in control (AHI < 15) and moderate-to-severe OSA (AHI > 15) patients. (**C**) Western blot analysis of PD-L1 expression on circulating sEVs from control (AHI < 15) and moderate-to-severe OSA (AHI > 15) patients. Expression of PD-L1 was normalized with total protein stained with Ponceau solution. (**D**) Correlation between PD-L1 expression on sEVs and AHI.

**Figure 3 ijms-25-03208-f003:**
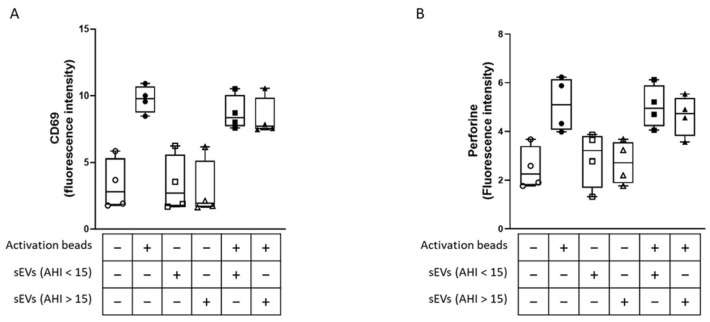
Circulating sEVs did not modify cytotoxic lymphocyte activation or cytotoxicity. Activation and cytotoxicity of CD8+ lymphocytes were analyzed by CD69 (**A**) and perforin (**B**) expression after incubation for 1 h with circulating sEVs from control (AHI < 15) and moderate-to-severe OSA (AHI > 15) patients, followed by stimulation with activation beads for 24 h (ratio beads-to-cell = 1:2). CD69 and perforin expression were analyzed by flow cytometry, and the results were expressed as fluorescence intensity.

**Table 1 ijms-25-03208-t001:** Demographic, respiratory, and metabolic characteristics of the study groups.

	AHI < 15	AHI > 15	*p*-Value
Subjects (n)	45	45	
Age (years)	42 [36–55]	58 [51–66]	<0.001
Body mass index (kg/m^−2^)	25.6 [23.5–29.0]	31.1 [27.3–35.0]	<0.001
Epworth sleepiness scale	11.0 [7.5–13.0]	9.0 [5.0–14.5]	0.180
AHI (events/h^−1^)	5.0 [2.5–8.5]	29.0 [22.0–43.0]	<0.001
Recording time with SpO_2_ < 90% (%)	0.0 [0.0–3.0]	37.0 [14.5–98.0]	<0.001
Mean nocturnal SpO_2_ (%)	94 [92–95]	92 [91–93]	<0.001
**Blood pressure (mmHg)**			
Systolic	122.0 [118.3–131.8]	132.0 [122.8–142.0]	0.003
Diastolic	70.00 [64.25–82.50]	83.00 [74.50–89.75]	<0.001
**White cell count (cells/mm^3^)**			
Neutrophils	3566 [2738–4298]	3790 [2763–4280]	0.780
Lymphocytes	145.2 [110.9–208.6]	146.8 [84.45–222.5]	0.589
Monocytes	512.6 [377.8–602.3]	575.8 [378.9–661.5]	0.365
Hemoglobin (g/dl^−1^)	14.65 [14.10–15.35]	14.90 [14.10–16.20]	0.988
Cholesterol (mM)	5.30 [4.50–6.08]	5.60 [4.80–6.10]	0.122
HDL-cholesterol (mM)	1.30 [1.10–1.58]	1.20 [1.10–1.40]	0.183
LDL-cholesterol (mM)	3.10 [2.28–3.68]	3.60 [2.70–4.10]	0.046
Triglycerides (mM)	1.40 [0.93–1.80]	1.70 [1.30–2.10]	0.209
Glucose (mM)	4.90 [4.63–5.28]	5.20 [4.90–5.80]	0.008

**Notes:** Data are presented as medians [IQR].

## Data Availability

Data is contained in the article and is available on request from corresponding author.

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
