# Peer review of "Small Extracellular Vesicles Harboring PD-L1 in Obstructive Sleep Apnea"

_ijms, 2024, doi:10.3390/ijms25063208_

Round 1
Reviewer 1 Report
Comments and Suggestions for Authors
Comments to the authors
Thank you for inviting me to review the manuscript entitled “Small extracellular vesicles harbouring PD-L1 in obstructive sleep apnea”. This is a case-control study where two groups of patients (moderate-severe OSA vs controls) have been compared in terms of the expression of PD-L1 on small extracellular vescicles and its role on lymphocyte activation and cytotoxicity.
This is a very-well written study, with a strong methodology (sample size calculation, justification of their cut-off values to categorize the patients) and clear results. Despite non-significant differences have been found, it is important to publish such studies, so that we do not incur in research bias which only publish positive findings. Moreover, the relationship between cancer and OSA is crucial and still an underexplored area. Many trials will probably be done with negative findings, until we finally clarify what the mechanism of action is that causes an increase tendency of cancer development among OSA patients.
Introduction: this is well-written. It provides the rational to the research question and the background which explain why the study is important to carry out as well as its justification.
Method: at first, I did not concur with the division of controls vs moderate-severe based on AHI of 15. However, the authors have provided a quite strong justification in their limitation sections. I may ask if possible to run the analysis just considering severe OSA patients (if the sample number is power enough for inferential statistics), as the authors mention that an increased tendency in cancer is seen among severe OSA patients.
Moreover, if the authors suggest that the time spent under 90% of saturation is more influential than AHI values, why didn’t the authors just classify their two groups based on the T90 instead of using the AHI? I anyway see that these data have been include in the baseline table, so the two values can correspond and overlap.
Line 230: it may be important to mention that a certified sleep physician read the PSG data and confirmed absence or presence of OSA. Was it the same sleep physician or it involved different sleep physicians?
Discussion: This is very well-written and exhaustive.
Conclusion: appropriate and sound.
Author Response
Dear Reviewer
Thank you for your reply concerning our manuscript ijms-2887654- Small extracellular vesicles harbouring PD-L1 in obstructive sleep apnea -. We are grateful to the reviewers for their time and effort in evaluating our paper, and appreciate their constructive comments and criticisms. The following is a point-by-point reply to your comments.
We hope that you will find this revised version suitable for publication in the International Journal of Molecular Sciences
Sincerely yours.
Dr Wojciech Trzepizur
- Method: at first, I did not concur with the division of controls vs moderate-severe based on AHI of 15. However, the authors have provided a quite strong justification in their limitation sections. I may ask if possible to run the analysis just considering severe OSA patients (if the sample number is power enough for inferential statistics), as the authors mention that an increased tendency in cancer is seen among severe OSA patients.
Moreover, if the authors suggest that the time spent under 90% of saturation is more influential than AHI values, why didn’t the authors just classify their two groups based on the T90 instead of using the AHI? I anyway see that these data have been include in the baseline table, so the two values can correspond and overlap.
Response: Those additional analysis were performed as suggested by the reviewer. It is now presented in the results section as follows: “No additional differences were observed using the AHI 30/h threshold (median [IQR] PD-L1/total protein 7.00 [2.60-8.85] for the AHI < 30/h group versus 5.8 [4.15-7.60] for severe OSA group; p>0.05) or separating patients by median T90 (median [IQR] PD-L1/total protein 8.00 [5.97-10.32] for the T90 ≤ 9% group versus 6.0 [4.65-7.80] for T90 > 9% group; p>0.05).”
- Line 230: it may be important to mention that a certified sleep physician read the PSG data and confirmed absence or presence of OSA. Was it the same sleep physician or it involved different sleep physicians?
Response: We apologize for the mistake in the previous version. All the recordings performed in this study were in laboratory PSG and read by certified physicians. This is now clearly exposed in the methods section as follows: “Patients were evaluated using in laboratory polysomnography (PSG) with the following criteria: apnea was defined as at least a 90% decrease in the oronasal sensor signal, and hypopnea was defined as at least a 30% decrease in the nasal pressure signal combined with either ≥ 3% arterial oxygen desaturation or an arousal (PSG), both lasting at least 10 seconds [38]. Certified sleep physicians read the PSG data and confirmed absence or presence of OSA and its severity. “

Reviewer 2 Report
Comments and Suggestions for Authors
The authors studied the characteristics of small extracellular vesicles: sEVs (concentration, size, cellular origin, and PDL1 expression) and their effects on CD8 lymphocytes activation and cytotoxicity in OSA patients. The authors did not find any differences concerning sEVs characteristics between subjects with AHI<15 and subjects with AHI>15. Moreover, sEVs in OSA patients have no significant effects on CD8 lymphocytes activation. The topic is interesting because most of the pathogenic mechanisms of cancers in a context of OSA remain unknown and the results of this study add to our knowledge some data that may be useful for future studies in the same field of research.
Below are some specific comments:
Introduction:
Page 1, line 30: the estimated prevalence of OSA in general population (%) was not reported.
Page 1-2; line 34-75: the authors presented a clear revue of literature on pathogenesis of cancer in OSA patients: they explained how intermittent hypoxia (IH) may lead to a dysregulation of immune response with inhibition of T cells activity. They also described how PD-1/PDL-1 pathway may be implied in this dysregulation in OSA patients as well as the possible participation of sEvs which characteristics and effects on immune response modulation are not established. Then the objectives of the study were clearly defined.
Materiels and methods:
Page 9; line 323: the authors reported the tests used for statistical analysis. However, the quantitative and qualitative variables were not defined; what are the parameters that were compared? These comparisons were made between which groups?... Before the paragraph “statistical analysis”, I recommend to add a data analysis part (with eventually a figure resuming the flowchart of the study) to describe all parameters studied and the comparisons that were made in the same order presented in part results.
Results:
Part Results was properly presented with text, table and figures
Discussion
Page 7; line 208-220: Among the limitations of the study, the type 3 sleep apnea testing which was used for sleep recording as an alternative for polysomnography may underestimate AHI: this was not mentioned by the authors.
Finally, I recommend the authors to discuss their findings regarding their usefulness and perspectives.
Author Response
Dear Reviewer
Thank you for your reply concerning our manuscript ijms-2887654- Small extracellular vesicles harbouring PD-L1 in obstructive sleep apnea -. We are grateful to the reviewers for their time and effort in evaluating our paper, and appreciate their constructive comments and criticisms. The following is a point-by-point reply to your comments.
We hope that you will find this revised version suitable for publication in the International Journal of Molecular Sciences
Sincerely yours.
Dr Wojciech Trzepizur
Reviewer 2:
- Introduction: Page 1, line 30: the estimated prevalence of OSA in general population (%) was not reported.
Response: the prevalence of OSA was added to the introduction section as follows: “With nearly one billion adults affected worldwide and a prevalence of 13% in men and 6% in women, obstructive sleep apnoea (OSA) syndrome is a highly prevalent disease characterized by recurrent episodes of complete or partial upper airway obstruction during sleep [1].”
- Page 9; line 323: the authors reported the tests used for statistical analysis. However, the quantitative and qualitative variables were not defined; what are the parameters that were compared? These comparisons were made between which groups?... Before the paragraph “statistical analysis”, I recommend to add a data analysis part (with eventually a figure resuming the flowchart of the study) to describe all parameters studied and the comparisons that were made in the same order presented in part results.
Response: No qualitative variables are finally included in the manuscript. The statistical section was adapted consecutively. A clarification of the analysis performed is now present in the statistical analysis section as follows: “Comparisons between groups based on AHI (no OSA or mild OSA [AHI<15/h; control group] versus moderate to severe OSA [AHI≥15/h]), were performed using Student or Mann-Whitney tests. The following data were successively compared: clinical, PSG and biological characteristics at inclusion; exosome size and concentration; exosome PD-L1 expression; lymphocyte CD69 and perforin expression in the presence of activating beads and exosomes. P<0.05 was considered to be statistically significant.”
- Page 7; line 208-220: Among the limitations of the study, the type 3 sleep apnea testing which was used for sleep recording as an alternative for polysomnography may underestimate AHI: this was not mentioned by the authors.
Response: We apologize for the mistake in the previous version. All the recordings performed in this study were in laboratory PSG. This is now clearly described in the methods section.
